# Structural Basis of a Novel Agonistic Anti-OX40 Antibody

**DOI:** 10.3390/biom12091209

**Published:** 2022-08-31

**Authors:** Jing Zhang, Xiaoyong Jiang, Han Gao, Fei Zhang, Xin Zhang, Aiwu Zhou, Ting Xu, Haiyan Cai

**Affiliations:** 1Key Laboratory of Cell Differentiation and Apoptosis of the Chinese Ministry of Education, School of Medicine, Shanghai Jiao Tong University, Shanghai 200025, China; 2Dingfu Biotarget Co., Ltd., Suzhou 215126, China

**Keywords:** OX40 antibody, crystal structure, agonistic, epitope specificity, cross-linking

## Abstract

Agonistic antibodies targeting co-stimulating receptor OX40 on T cells are considered as important as (or complementary to) the immune checkpoint blockers in cancer treatment. However, none of these agonistic antibodies have reached the late stage of clinical development partially due to the lack of intrinsic potency with the correlation between binding epitope and activity of the antibody not well understood. Here, we identified a novel anti-OX40 agonistic antibody DF004, which stimulated the proliferation of human CD4^+^ T cells in vitro and inhibited tumor growth in a mouse model. Our crystallography structural studies showed that DF004 binds to the CRD2 region of OX40 while RG7888, an OX40 agonist antibody developed by Roche, binds to CRD3 of OX40 to the diametrically opposite position of DF004. This suggests that the agonistic activities of the antibodies are not necessarily epitope dependent. As their agonistic activities critically depend on clustering or cross-linking, our structural modeling indicates that the agonistic activity requires the optimal positioning of three Fc receptor/antibody/OX40 complexes on the cell membrane to facilitate the formation of one intracellular hexameric TRAF complex for downstream signal transduction, which is relatively inefficient. This may explain the lack of sufficient potency of these OX40 antibodies in a therapeutic setting and sheds light on the development of cross-linking-independent agonistic antibodies.

## 1. Introduction

Immune checkpoints consist of both stimulatory and inhibitory receptors that regulate immune homeostasis and autoimmunity [1,2,3]. Blocking antibodies targeting these immune checkpoint receptors can stimulate antitumor immunity and have achieved great success in cancer therapeutics in recent years [4,5,6]. Currently, PD-1/PD-L1 blocking antibodies have been approved for several types of tumors, demonstrating the importance of the immune system in combating cancer [7]. However only about 10–30% of the cancer patients responded to the PD-1/PD-L1 treatment, it is important to identify new immuno-therapeutic targets to meet the clinical needs [1,8].

Agonistic monoclonal antibody/antibodies (mAb) targeting co-stimulatory receptors have been under intensive clinical development for cancer therapy, in particular, tumor necrosis factor receptors (TNFR) superfamily members, such as CD40, 4-1BB, and OX40 [9]. TNFR members are characterized by an extracellular domain composed of several cysteine-rich domains (CRDs) for interaction with their respective homo-trimeric, tumor necrosis factor (TNF)-like ligands [10]. OX40, also known as TNFRSF4 or CD134, is highly expressed on activated effector and memory T cells typically between 24 and 96 h after cognate antigen recognition [11,12,13]. OX40L, as the only ligand of OX40, is mainly expressed on activated antigen-presenting cells (APCs) [14]. Upon OX40-OX40L interaction, downstream intracellular pathways in T cells are activated, such as nuclear factor-κB (NF-κB) and nuclear factor of activated T-cells (NF-AT), leading to the activation of CD4^+^ and CD8^+^ T cells and the counteraction of the suppression by regulatory T cells (Tregs) [15]. Moreover, high OX40 expression in the tumor immune infiltration is associated with a favorable prognosis for several types of cancer [15,16]. Therefore, OX40 is widely recognized as one of the most promising targets for novel cancer immunotherapy.

Several agonistic antibodies targeting OX40 have been developed and evaluated in early clinical trials [17,18,19,20]. For example, RG7888 (MOXR0916/pogalizumab), a humanized effector-competent OX40 agonist antibody developed by Roche, has completed the preliminary toxicity test in combination with atezolizumab in patients with advanced solid malignancies; results showed that the combination was well tolerated with no dose-limiting toxicities [19]. Other OX40 agonist antibodies, such as MEDI-0562 (developed by AstraZeneca) and 11D4 (PF-04518600, developed by Pfizer), were also evaluated in phase I clinical trials [19,20]. Despite the significant advancement of a large pipeline of OX40 agonist antibodies in development, none have reached late-stage clinical trials, and several have been withdrawn [21].

The lack of intrinsic potency is one of the most important reasons, and it remains a question as to whether an OX40 antibody is capable of sufficiently driving receptor signaling in a therapeutic setting [22]. The other issue in developing OX40 agonist antibodies is how the epitopes on OX40 targeted by the antibody would influence the type and strength of its effector functions [23,24]. For anti-CD40 mAb, the membrane distal CRD1-binding mAbs were shown to be strong agonists of CD40 with the membrane-proximal mAb less potent [24]. Furthermore, mAb-binding CRD2-4 blocked CD40L and were potent antagonists. In contrast, Zhang et al. reported that mAb binding to mouse OX40, which blocked ligand-binding and bound CRD2, or bound at the membrane-proximal domain (CRD4), provided stronger agonistic and anti-tumor activity than mAbs binding CRD1 and CRD3 [25]. Interestingly, Griffiths et al. examined a panel of anti-hOX40 antibodies and showed that both the isotype and epitope of the antibodies have significant impacts on therapy with membrane-proximal mAb delivering more powerful agonisms [26]. Therefore, it is somewhat confusing how the binding epitope would correlate with the agonistic activity of an antibody.

To address this, here, we identified and characterized an anti-OX40 antibody with agonistic activity and solved the crystal structure of its complex with human OX40. As a comparison, we also solved the crystal structure of the OX40 complexed with an agonistic antibody RG7888 developed by Roche. The structural analysis showed that both agonistic antibodies block the binding of OX40 with OX40L through binding to distinctively different epitopes on OX40. Both require optimal positioning of three Fc receptor/antibody/OX40 complexes on the cell membrane to facilitate the downstream signal transduction, which is relatively inefficient or less potent. This may explain the lack of intrinsic potency of some anti-OX40 antibodies in therapy.

## 2. Materials and Methods

### 2.1. Reagents

The HEK293F cells for protein expression were purchased from Thermo Fisher (Waltham, MA, USA). The 293T cells were obtained from ATCC. TRIzol reagent was obtained from Thermo Fisher Life Technologies (Carlsbad, CA, USA). SuperScript IV First-Strand Synthesis kit was purchased from Thermo Fisher Scientific Baltics UAB (Graiciuno, Vilnius, Lithuania). Polyethylenimine (PEI) was purchased from Polysciences (Warrington, PA, USA). The affinity HisTrap column, ion-exchange column, HiLoad 16/600 Superdex 200 pg gel filtration column, and protein A column were all purchased from GE Healthcare (Marlborough, MA, USA). The antibody goat (Fab’)_2_ anti-human IgG Fc (HRP) was purchased from Sigma (Saint Louis, MO, USA, no. A0170). The luciferase activity assay system was purchased from Promega (Madison, WI, USA, E6110). CFSE (no. 422701), anti-CD3 antibody (no. 300414) and anti-hCD28 antibody (no. 302913) were all purchased from BioLegend (San Diego, CA, USA).

### 2.2. Antibody Generation

Peripheral blood mononuclear cells (PBMCs) were isolated from 50 healthy donors by standard density gradient centrifugation. RNA was isolated using the TRIZOL reagent. cDNA was synthesized using the SuperScript IV First-Strand Synthesis kit primed with oligo (dT) and subsequently used to amplify the variable heavy chain (VH) and variable light chain (VL) repertoire. The single-chain antibody fragment (scFv) was obtained by overlapping PCR with VH and VL. ScFv libraries were constructed by co-transforming the linearized pFab vector and the PCR product containing homology arms by yeast gap repair. The pFab vector system allows the display of fusion proteins containing a scFv followed by Aga2p and a Myc tag. The transformation of EBY100 with library plasmids was performed using a standard electroporation method [27]. Expression-positive yeast cells were sorted by magnetic-activated cell sorting (MACS) (Miltenyi Biotec, Bergisch Gladbach, Germany) and then by fluorescence-activated cell sorting (FACS) (BD, Aria III, Franklin Lakes, NJ, USA) using biotinylated hOX40. Individual clones were picked for sequencing.

### 2.3. Protein Expression and Purification

Recombinant human OX40 protein and antibodies were prepared through transit expression in HEK293F cells using a similar procedure as previously described [28]. Briefly, recombinant OX40 with a Histag at the C-terminus was purified by an affinity HisTrap column and then by a HiLoad 16/600 Superdex 200 pg gel filtration column in 20 mM Tris-HCl pH 7.4, 150 mM NaCl. The cDNAs encoding the heavy chain (VH-CH1) with an 8×His tag and light chain (VL-CL) of DF004-Fab or RG7888-Fab were cloned into pCEP4 expression vectors. They were co-transfected into 293F suspension cells using polyethylenimine (PEI). The secreted Fab fragment of the antibody was purified by a Ni-NTA column (Roche, Basel, Switzerland) and the eluted samples were further purified by an ion-exchange column. The DF004-Fab/OX40 and RG7888-Fab/OX40 complexes were obtained by mixing the corresponding protein components at equimolar ratios and further purified using a HiLoad 16/600 Superdex 200 pg column. Purified complexes were concentrated to ~15 mg/mL for crystallization. The full antibodies (DF004, RG7888, 11D4, and MEDI-0562) from the expression medium were purified by a protein A column.

### 2.4. BLI Assay

Binding kinetics of antibody DF004 to OX40 variants were determined using FortéBio Octet K2 instrument. Protein A sensors were activated in phosphate-buffered saline with 0.1% bovine serum albumin (BSA) by agitating 96-well microtiter plates at 1000 rpm to minimize nonspecific interactions. The final volume for all solutions was 200 μL per well. The sensors were loaded with 10 μg/mL DF004 for 40 s before equilibration for 60 s in phosphate-buffered saline (PBS) with 1% BSA. Variants of OX40 were prepared as a two-fold serial dilution in the same buffer and incubated with sensors for 120 s. Then, OX40 variants bound on the sensors were allowed to dissociate for up to 320 s depending on the observed dissociation rate. All measurements were corrected for baseline drift by subtracting a control sensor exposed to the running buffer. Data analysis and curve fitting were carried out using Octet software.

### 2.5. Enzyme-Linked Immunosorbent Assay (ELISA) Assay

Human OX40 variants were coated onto a 96-well plate overnight at 4 °C. The plate was washed and blocked for 2 h in PBS containing 3% BSA at 37 °C. Serially diluted antibodies were added to the plate in duplicates and incubated for 2 h at 37 °C. After washing, goat (Fab’)_2_ anti-human IgG Fc (HRP) was added and incubated at 37 °C for 1 h. Plates were washed with signal developed using 3,3′,5,5′-tetra-methylbenzidine substrate solution (Solarbio, Beijing, China), according to the manufacturer’s instructions. Absorbance at 450 nm was measured on a Molecular Devices SpectraMax i3 system with SoftMax Pro software.

### 2.6. Luciferase Reporter Assay

The 293T cells expressing human OX40 in pcDNA3.1 (+) vector and NF-κB-luciferase constructed in pf9a (Promega, Madison, WI, USA) were cultured in the 96-well plate (2.5 × 10^4^ cells per well) at 37 °C. In the assay, cells were incubated with different concentrations of anti-OX40 antibodies for 6 h. Antibodies were cross-linked using peroxidase-conjugated goat anti-human IgG Fc polyclonal antibody. The luciferase activity was detected according to the manufacturer’s recommendation (Promega: E6110).

### 2.7. CD4^+^ T Cell Proliferation Assay

Human PBMCs were isolated from the peripheral blood of healthy donors by standard density gradient centrifugation. Naive CD4^+^ T cells were enriched according to the manufacturer’s recommendation (stem cell: 17952). Then, CD4^+^ T cells were labeled with 5 μM CFSE and incubated at 37 °C for 20 min. After being washed with PBS twice, CD4^+^ T cells were seeded in an anti-CD3 (3 μg/mL) antibody-coated 96-well plate. At the same time, cells were incubated with serial dilutions of anti-OX40 antibodies or 10 μg/mL hIgG for 5 days. Anti-CD28 antibodies (1 μg/mL) were added as a positive control. The proliferation of CD4^+^ T cell was assessed by multicolor-flow cytometry.

### 2.8. MC38 Tumor Model in Mice

To evaluate the antitumor effect of DF004 in vivo, a xenograft mouse model was prepared by implanting subcutaneously (SC.) MC38 cells (5 × 10^5^ cells in PBS) into the right flanks of hOX40 knock-in female C57BL/6 mice. On day 6 post-implantation, mice were randomized into 4 groups (N = 6) with a mean tumor volume of approximately 107 mm^3^. On days 7, 10, 14, 17, 21 and 25 post-implantation, mice were injected intraperitoneally with DF004 or control IgG. Tumor volume and body weight were measured twice a week. This study was performed in strict accordance with institutional guidelines and approved by the Institutional Animal Care and Use Committee in Shanghai Research Center for Model Organisms and the IACUC permit number was 2018-0004.

### 2.9. Structure Determination

Crystallization trials were performed at room temperature using the sitting drop vapor diffusion method by mixing equal volumes of complex and reservoir using a TTP LabTech Mosquito robot and commercially available screens. After optimization of initial hits, the OX40/DF004-Fab complex was crystallized at 15 mg/mL in 0.2 M ammonium dihydrogen phosphate, 20% PEG 3350. Crystals were cryoprotected in the same conditions plus 25% glycerol in the mother liquor. OX40/RG7888-Fab complex was crystallized at 0.1 M potassium sodium tartrate, 14% PEG 3350, and 10% glycerol. Then, all crystals were flash-cooled in liquid nitrogen. Diffraction data were collected on beamlines BL17U1 and BL19U1 of Shanghai Synchrotron Radiation Facility (SSRF), China [29]. The data were indexed and processed with iMosflm and scaled with Aimless from the CCP4 suite [30]. The initial phases were obtained by molecular replacement using Phaser [31] with models from PDB 2HEV [32] and PDB 6OKM [22]. The models were subsequently built using Coot [33] and refined using Refmac [34]. All structural figures were generated using PyMOL software [35]. Crystal structural data have been deposited in the Protein Data Bank with accession codes PDB: 8AG1 (OX40/DF004-Fab) and PDB: 7YK4 (OX40/RG7888-Fab).

### 2.10. Quantification and Statistical Analysis

Flow cytometry data were analyzed with FlowJo v10.4. Statistical analyses were done with Prism 8 (GraphPad Software). *P* values were calculated with the unpaired two-tailed Student’s *t*-test or two-way ANOVA where indicated (ns, not significant; *, *p* < 0.05; **, *p* < 0.01).

## 3. Results

### 3.1. Generation and Characterization of An OX40 Antibody DF004

A few hundred clones targeting the OX40 extracellular segment were generated from the antibody screening platform. After several rounds of primary screening based on binding affinity, one of the clones with high affinity towards OX40 was selected with variable fragments grafted onto IgG1 Fc. This antibody was termed DF004 and it binds human OX40 (hOX40) with a Kd value of 4.39 nM when measured by biolayer interferometry technology (BLI) (Figure 1A).

To evaluate the effects of DF004 on T-cell activation in vitro, we established a 293T-hOX40-NF-κB stable cell line. This luciferase reporter assay revealed that DF004 and all other antibodies including RG7888, 11D4, and MEDI-0562 (Figure 1B) have relatively weak activities in activating the NF-κB pathway by themselves. As it is known that the cross-linking of IgG Fc by its receptors is often crucial for activating the downstream signaling pathways [36,37], we further assessed the effect on NF-κB activation in the presence of an anti-Fc antibody for cross-linking. This showed a significant increase (about 5-fold) in NF-κB activation signal from all the antibodies tested. DF004 and RG7888 have similar agonistic activity, which is higher than those of 11D4 and MEDI-0562 (Figure 1B). The above results demonstrated the critical role of cross-linking in amplifying the antibody-mediated downstream activation signal.

As OX40 co-stimulation induces profound CD4^+^ and CD8^+^ T cell proliferation, differentiation, and survival [38], DF004 and other OX40 antibodies were then tested for their effects on primary human CD4^+^ T cells. As shown in Figure 1C, DF004, RG7888, and MEDI-0562 had comparable and relatively strong activity in stimulating the proliferation of CD4^+^ T cell dose-dependently, while the activity of 11D4 appeared slightly weaker (Figure 1C). Subsequently, we assessed the anti-tumor efficacy of DF004 using the humanized C57BL/6 mouse model bearing MC38 tumors with DF004 administered at three doses (0.3 mg/kg, 1.0 mg/kg, and 5.0 mg/kg). The results demonstrated that DF004 significantly inhibited tumor growth in a dose-dependent manner (Figure 1D). Bodyweight changes among the groups were insignificant during the treatment period (Figure 1E). The in vivo activity of DF004 is comparable to the previous characterization of other anti-OX40 agonist antibodies [18]. Overall, these results confirmed that DF004 is an agonistic antibody targeting OX40.

### 3.2. Structure of OX40/DF004-Fab Complex

In order to understand the structural basis of the agonistic DF004, we then determined the crystal structure of DF004-Fab complexed with the extracellular fragment of hOX40 at 3.3 Å resolution (Appendix A). There are two copies of DF004-Fab/OX40 complexes in the asymmetric unit (Figure 2A). Although the extracellular ligand-binding domain of OX40 is composed of four cysteine-rich domains (CRDs) based on the number of cysteine and topology of the cysteine connectivity [39,40], only the first three CRDs (CRD1,2,3) were resolved in the electron density and CRD4 was not built in the final model (Figure 2B). The structure clearly showed that DF004 is mainly bound to the CRD2 domain of OX40 through its HCDR2 (complementarity-determining region) and the HCDR3 loop of the heavy chain (VH) and LCDR3 loop of the light chain (LH). The binding interface has a burial surface area of 1156 Å^2^ (Figure 2B) and is formed through both hydrophobic and hydrophilic interactions between DF004 and OX40 (Appendix A). Arg52, Asn102 of the HCDR3 loop, and Tyr98 of the LCDR3 loop in DF004 form polar interactions with the main chain of Pro69 from CRD2 of OX40. Arg91 of the LCDR3 loop formed a hydrogen bond with the main chain of Gly70 in the CRD2 region of OX40 (Figure 2C). Trp94 and Tyr98 in the LCDR3 loop of the DF004 pack with Phe71 of OX40 formed potential π–π hydrophobic interactions (Figure 2D). Notably, the P^69^G^70^F^71^ loop from CRD2 of OX40 was docked in the surface cavity formed by VH and VL of DF004 (Figure 2E).

To assess the contribution of these OX40 residues towards DF004 binding, they were mutated to alanines with their binding affinities measured by BLI (Figure 2F and Appendix A). The binding affinity of OX40-G70A towards DF004 decreased by nearly 32-fold (kd value) (Figure 2F), which indicated close packing of this loop on DF004 with an extra CH3 group introduced from Gly-to-Ala mutation leading to substantial loss of the binding interactions. Notably, no binding of the OX40-G70A variant towards DF004 could be detected by BLI (Appendix A). The binding activity of these OX40 variants toward DF004 was further evaluated by ELISA. DF004 bound the wild type OX40 with an EC_50_ value of ~0.09 nM (Figure 2G), while no DF004 binding on OX40-F71A could be detected even at 10 ug/mL (~66.7 nM), indicating the critical role of Phe71 in the DF004/OX40 binding interface (Figure 2G). Mutation of other residues of OX40, such as Arg65, Pro83, and Trp86 involved in the interaction with DF004 had little impact on the binding between OX40 and DF004 (Appendix A). Overall, these mutagenesis studies highlighted the critical role of the P^69^GF^71^ loop of OX40 in the binding of DF004. Furthermore, sequence alignment of the CDR2 domains from human, mouse, and rat OX40 demonstrated that the key residues Gly70 and Phe71 are conserved (Appendix A). This indicates the potential cross-reactivity of DF004 towards the mouse or rat OX40, and DF004 might be a good candidate for direct evaluation of its antitumor activity and adverse events in animal models.

### 3.3. Comparison of OX40-Antibody Complex Structures

RG7888 is a well-studied OX40 agonist antibody for its in vitro activities and in vivo studies in mice models [18]. To compare the epitope of DF004 and RG7888, we then solved the crystal structure of RG7888-Fab complexed with OX40 at 2.7 Å resolution. The C-terminal part of OX40 containing CRD2-4 was resolved in the final model of the complex while CRD1 was not built due to poor electron density (Figure 3A). The structure showed that the binding site of RG7888 on OX40 was located on the CRD3 domain of OX40. RG7888 formed several hydrogen bonds with OX40, involving four RG7888 residues (Tyr32, Tyr49, Arg53, and Tyr103) and five OX40 residues (Ser91, Asp124, Cys125, Pro127, and Cys128) (Figure 3B and Appendix A). The buried interface surface area is about 974 Å^2^ with 559 Å^2^ from the heavy chain and 415 Å^2^ from the light chain of RG7888. These structural features are consistent with previous characterizations [22].

The overall configuration of OX40 from the RG7888-OX40 and DF004-OX40 complex is similar to that seen in the OX40/OX40L complex (PDB code: 2HEV [32]) (Figure 3C). However, DF004 and RG7888 bind OX40 surface on diametrically opposite positions with DF004 on CRD2 and RG7888 on CRD3 (Figure 3C,D). The other crystal structure available in the protein data bank (PDB) is also an agonistic anti-OX40 antibody (3C8) developed by Roche [22], which is also bound on CRD2 of OX40. Notably, 3C8 bound OX40 in a different orientation to DF004 on the OX40 surface (Figure 3C,D). A detailed structural analysis of all these available crystal structures of OX40-antibody complexes revealed that both the heavy and light chains of each antibody contributed nearly equal surface area toward the binding of OX40 (Figure 3D and Appendix A). All three antibodies covered ~1000 Å^2^ surface areas of OX40, which are significantly larger than that of OX40L (~400 Å^2^), which is consistent with the higher affinities of the antibodies towards OX40. Notably, OX40L binding covers CRD1-3, DF004 covers CRD1-2, RG7888 covers CRD2-3, while 3C8 covers solely CRD3 (Figure 3E). Overall, all three agonistic OX40 antibodies bind OX40 with some common features, such as high affinity and total surface areas covered (but they do have some distinctly different features).

### 3.4. Mechanism of the Agonistic Activities

It is known that OX40L, the ligand for OX40 (CD40L/CD154) is trimeric and initiates OX40 activation by forming hexameric OX40/OX40L complexes on the cell membrane [32]. Subsequently, the triangularly positioned tails of OX40 (about 51 Å away from each other) would recruit three TRAFs to form a hexameric intracellular TRAF complex leading to immune activation (Figure 4A). The relative positions of OX40 molecules on the cell membrane would play a key role in initiating the downstream signal transduction. Structural alignment of the DF004/OX40 or RG7888/OX40 complex with that of OX40/OX40L showed that the binding of both antibodies would block the hexameric OX40/OX40L complex formation (Figure 4B,C). From the perspective of preventing OX40L from activating OX40, both antibodies could be regarded as antagonists. However, the binding of the antibody on OX40 in the cell membrane would have a crowding effect which effectively makes OX40 ‘bulkier’. This could lead to an increase in the relative concentration of OX40 on the cell membrane and subsequent downstream signal transduction. Cross-linking of the antibody by their Fc domains would significantly increase the local concentrations of OX40 for TRAF trimerization and downstream signal transduction. Structural modeling of a full antibody complexed with OX40 showed that the distance between the two OX40 molecules is likely more than 100–130 Å apart due to the constraints of the disulfide bond at the hinge region of IgG [41,42] (Figure 4D,E). Clearly, the two OX40 molecules from a full antibody are unlikely to be involved in the same TRAF hexameric complex. Therefore, in experimental conditions where OX40L is abundantly present, the antibody would mainly function as an antagonistic agent; however, in an environment where Fc receptors on the cell surface are abundant or in the presence of anti-Fc cross-linkers, these antibodies will function agonistically as shown in our model (Figure 4F). The overall effect of these antibodies will critically depend on the experimental setup reflecting the combined antagonistic and agonistic activities.

## 4. Discussion

One strategy to promote an antitumor immune response that is different from checkpoint inhibition is to activate the TNFR superfamily of costimulatory T-cell receptors [19]. Agonistic mAbs targeting TNFRs could activate stimulatory signaling via cross-linking of co-stimulatory molecules, which is similar to that of receptor oligomerization upon ligand binding [45,46]. OX40 is one of the most promising targets where the agonistic activities of anti-OX40 antibodies tested clinically are often mediated by the in vivo Fc receptors. Some of the agonistic antibodies were optimized via Fc engineering strategies either to facilitate receptor clustering on the cell surface directly or to enhance binding affinity with FcγRIIB [47,48]. Nevertheless, none of the OX40 agonistic antibodies have progressed to late-stage clinical trials. The major hurdle in the development of the agonistic OX40 antibody is associated with either potential toxicity or a lack of intrinsic potency [49,50]. These fundamental issues likely lie with the OX40 surface region targeted by the antibody and other antibody related-interactions [23,24] which could only be addressed by detailed structural studies and thorough in vivo analysis under well-controlled experimental settings.

Previous studies have demonstrated that the epitope of the antibody is associated with different properties such as potency and therapeutic index [51] while antibodies binding on similar areas of CD40 or OX40 have shown inconsistent agonistic activities [24,25,26]. Here we have identified a novel anti-OX40 antibody DF004, which could induce NF-κB signaling and promote T-cell activation, survival, and expansion in vitro. Animal experiments showed effective inhibition of DF004 on MC38 tumor using humanized OX40 mice when administrated as a single agent. Additionally, the crystal structures of DF004 and another agonistic antibody RG7888 revealed that they bind to different surface areas of OX40 with both blocking the formation of signal transducing OX40L/OX40 complex. Since these antibodies have similar agonistic activities, this suggests the epitope of the antibody is not directly correlated with the activity of the antibody. In other words, it is not necessarily true that antibodies bound to a particular surface area on OX40 will have the same agonistic activities.

Furthermore, our structural modeling of the complex of OX40 with a full antibody DF004 or RG7888 on the cell membrane indicates that the relative positions of Fc domains of the antibodies are different. This may lead to differences in the engagement of cell surface Fc receptors leading to different distributions of OX40 along the cell surface and ultimately different efficiency in activating downstream signal pathways under physiological conditions. When considering different types and abundance of Fc receptors on the cell surface, the isotypes of antibodies, the presence of nature ligands, etc., it becomes even more challenging in predicting the activity from the binding epitopes. So, the development of antibodies targeting co-stimulatory TNFRs likely requires studies on a case-by-case basis [25]. Most importantly, our structural modeling of the full antibody complex indicates that the agonistic antibodies characterized here require three spatially positioned Fc receptor/antibody/OX40 complexes on the cell membrane to generate one TRAF complex for downstream signal, which is likely inefficient or less potent during therapy. Therefore, the development of cross-linking-independent agonistic antibodies with less toxicity may provide better therapeutic potential.

## Figures and Tables

**Figure 1 biomolecules-12-01209-f001:**
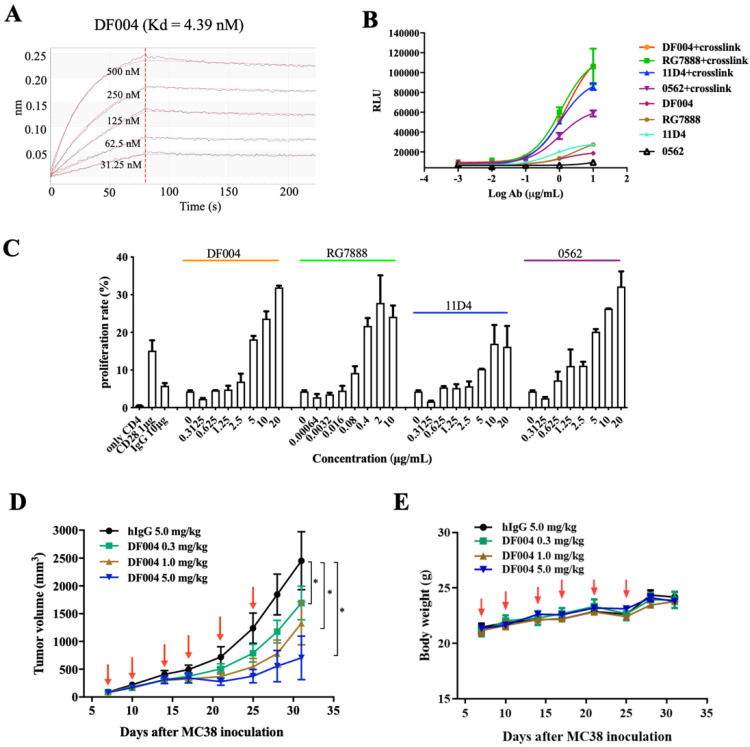
Characterization of an OX40 antibody DF004. (**A**) Binding affinity of DF004 towards the extracellular domain of OX40 assessed by biolayer interferometry with immobilized DF004 on the sensor. (**B**) Activation of OX40-NF-κB pathway by DF004, RG7888, 11D4, and 0562 (MEDI-0562) evaluated by a Luciferase reporter assay in the absence or presence of extrinsic cross-linkers. The x-axis represents the logarithmic concentrations of each antibody. (**C**) The proliferation of CD4^+^ T cells induced by DF004, RG7888, 11D4, and 0562 (MEDI-0562). (**D**) Tumor growth curve of human OX40 knock-in mice bearing MC38 tumors treated with DF004 at 0.3 mg/kg, 1.0 mg/kg, and 5.0 mg/kg, respectively. DF004 was administrated by intraperitoneal injection at 7, 10, 14, 17, 21 and 25 days after MC38 cell implantation. The asterisks indicate statistical compared to hIgG group (unpaired two-tailed Student’s t-test, * *p* ≤ 0.05). (**E**) The body weights of the mice were measured during the experiment.

**Figure 2 biomolecules-12-01209-f002:**
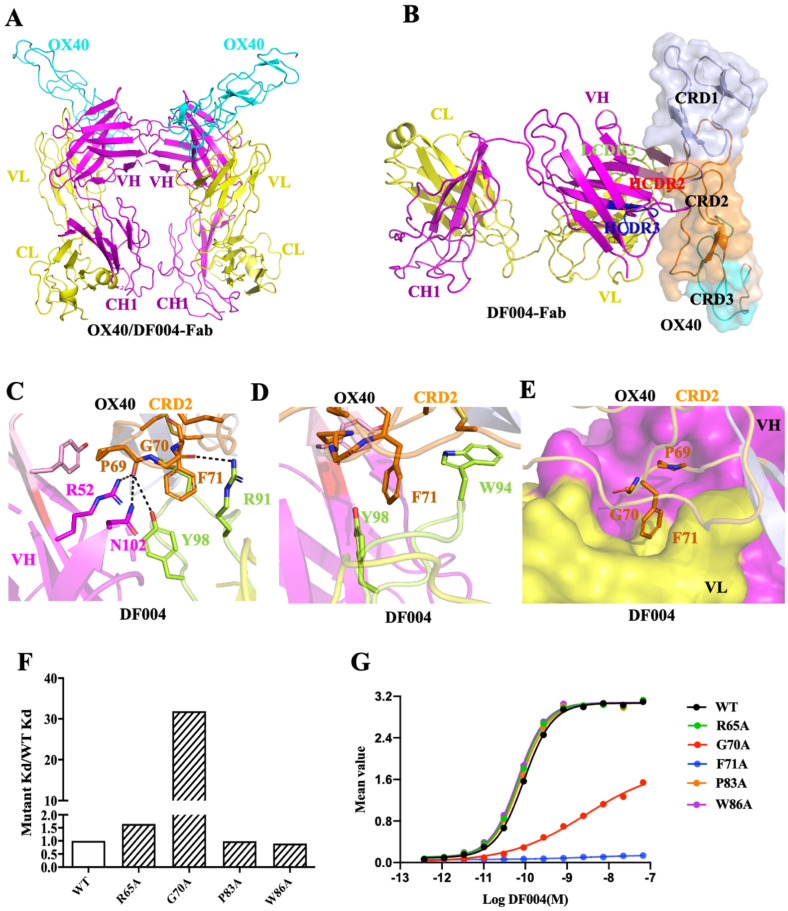
Structural analysis of OX40/DF004-Fab complex. (**A**) Two DF004-Fab/OX40 complexes in the asymmetric unit. VH-CH1 and VL-CL of DF004 are colored in magenta and yellow, respectively. OX40 is shown in cyan. (**B**) DF004 is mainly bound on the CRD2 domain (orange) of OX40 (semi-transparent surface). (**C**,**D**) Polar and hydrophobic interactions between OX40 and DF004-Fab. (**E**) The P^69^G^70^F^71^ loop of OX40 is inserted in the cavity formed by VH and VL of DF004. OX40 residues involved in the interaction are shown as orange sticks, and residues from HCDR3 and LCDR3 of DF004 are colored in magenta and green, respectively. (**F**) Binding affinity changes of OX40 variants. Notably, no signal could be detected from the F71A variant indicating a significant loss of binding affinity (Appendix A). (**G**) Binding affinity of OX40 variants assessed by ELISA. Serially diluted (3x) hOX40 variants were coated on a 96-well plate and then DF004 was added with the bound DF004 detected by goat (Fab’)2 anti-human IgG Fc (HRP). Similar results were obtained from two independent experiments with one of them shown here. Abbreviation: the first constant region of heavy chain 1 (CH1), constant region of light chain (CL), variable region of light chain (VL), variable region of heavy chain (VH), cysteine-rich domain (CRD).

**Figure 3 biomolecules-12-01209-f003:**
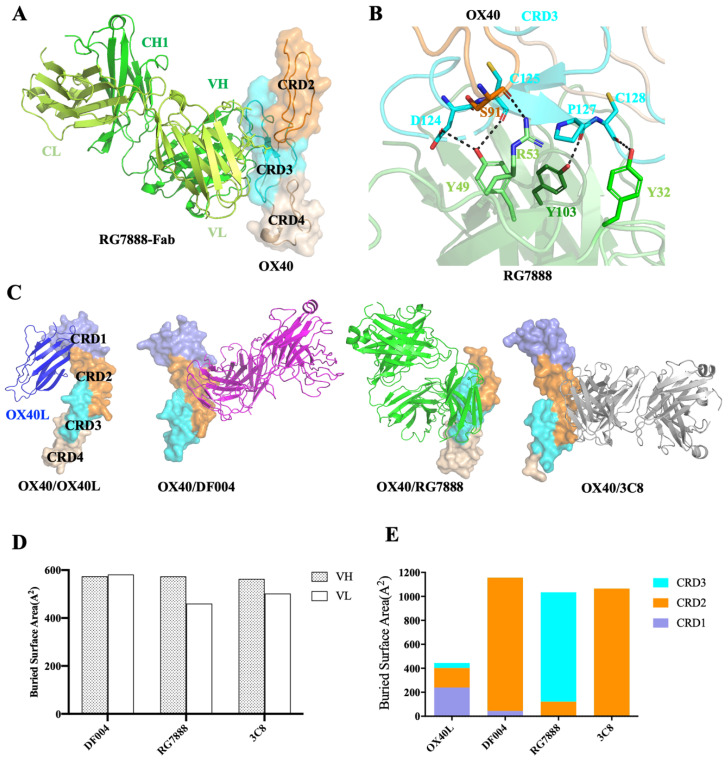
Structure analysis of the OX40 complexes. (**A**) RG7888 is mainly bound on the CRD3 domain of OX40. (**B**) Interactions between OX40 and RG7888. Residues involved in RG7888-OX40 interface are shown as sticks. Dashed lines represent the polar interactions. (**C**) Structures of the OX40 complexed with OX40L (blue, PDB 2HEV), DF004-Fab (magentas), RG7888-Fab (green) or 3C8 (gray, PDB 6OKM) with OX40 shown as colored surface (CDR1: slate; CRD2: orange; CRD3: cyan; CRD4: wheat). (**D**) Binding surface areas contributed by the VH or VL domain of the antibodies DF004, RG7888 and 3C8. (**E**) The surface areas of CRDs from OX40 covered by OX40L, DF004, RG7888 and 3C8. Surface area values in (**D**,**E**) were calculated by PISA (PDBePISA).

**Figure 4 biomolecules-12-01209-f004:**
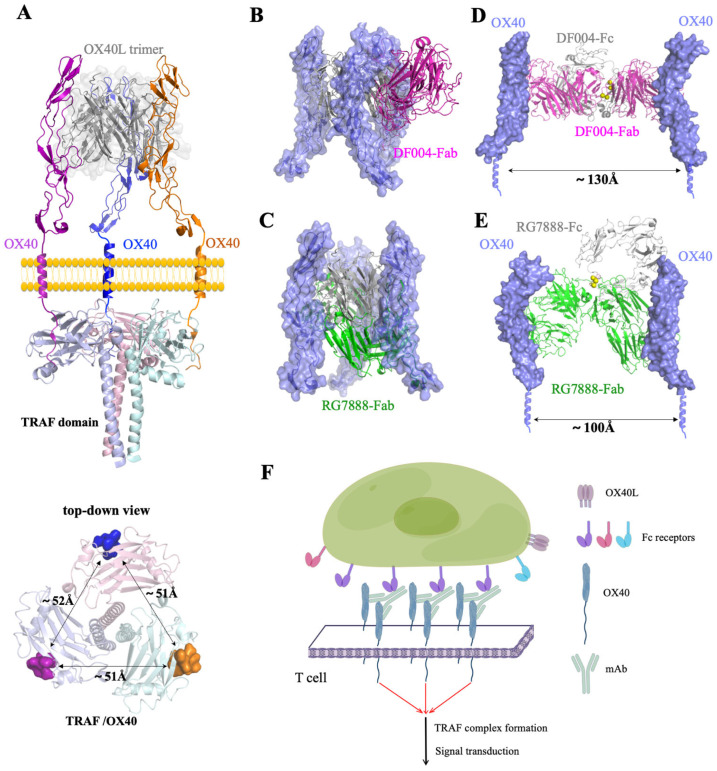
Mechanism of antibody-mediated activation of OX40. (**A**) OX40 activation by forming hexameric OX40/OX40L complexes on the cell membrane. This allows optimal positioning of three intracellular tails of OX40 to recruit three TRAFs to form a hexameric intracellular complex leading to immune activation. Three OX40 molecules are colored purple, blue, and orange, respectively. The model of this OX40 complex is derived from the previous study of TRAF hexameric complexed (PDB 1CZZ) [43]. The top-down view showed the triangularly positioned intracellular tails of OX40 which are ~51 Å apart. Overlaid structures of DF004/OX40 (**B**) or RG7888/OX40 (**C**) with the OX40/OX40L hexamer showing the binding of either antibody on OX40 would block the hexameric complex formation. A model of DF004 (**D**) or RG7888 (**E**) full antibody (IgG1 isotope, PDB 1HZH [44]) complexed with OX40 on the cell membrane showing the distance between the two OX40 is likely more than 100 Å or 130 Å apart due to constraints from the hinge disulfide bonds (yellow spheres). (**F**) An illustration showing Fc receptors mediated OX40 activation by antibody DF004 or RG7888 (by Figdraw). Localized Fc receptors on the immune cell surface would engage with Fc domains of anti-OX40 antibodies and allow clustering of intracellular tails of OX40. Notably, activation requires three optimal positioned OX40 tails (~51 Å apart) from three antibody complexes for the formation of one intracellular hexameric TRAF complex formation.

## Data Availability

Not applicable.

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
