# Peer review of "Structural Basis of a Novel Agonistic Anti-OX40 Antibody"

_biomolecules, 2022, doi:10.3390/biom12091209_

Round 1

Reviewer 1 Report

Summary: 

Zhang and colleagues here report on a new agonistic but blocking antibody DF004 targeting OX40. They compare its binding characteristics and functional efficacy with another stimulating anti-OX40 RG7888 antibody. The authors where able to characterize binding and affinity of their novel antibody and show induction of intracellular signaling, T cell proliferation and anti-tumor efficacy in the MC38 humanized mouse model. The state of the art literature has vastly been included and adequately discussed with their new findings.

Major Comments:

- The authors should start the introduction on checkpoint molecules with state of the art references and thus being currently published (lines 30-36)

- lines 182-185, the authors compare BLI and ELISA in order to assess the affinity of DF004 to human OX40. It would be helpful for the reader to appreciate the data better, if EC50 was also indicated in molar concentration.

- line 185, results cannot be discussed before they have been described. these ELISA data has to be part of the discussion section and not results

- the authors need to provide a reagents section and also a description of the human samples used. informed consent etc, ethic board approval...

- would be helpful to describe the yeast display a bit more in detail

Minor Comments:

- line 305 is also discussion

- need to provide clone numbers to functional antibodies, e.g. what was the anti-CD3 clone used?

Reviewer 2 Report

Zhang et al. have studied the binding mode and kinetics of antibody DF004 on OX40 (a important target for cancer therapy) receptor by crystallographic (protein crystal structures), biochemical (ELISA, BLI and Luciferase reporter assay) and in vivo/in vitro (Mice model and cell lines) analysis. The binding mode of DF004 is compared with previously known antibody RG7888. The DF004 is found to bind different site (CRD2 region) than RG7888 (CRD3 region) on OX40 receptor Overall, the manuscript is well-written, methodology is well-designed, results are well-described and interpreted, and the manuscript reveals important and novel scientific information. The manuscript can be accepted for publication after revision as detailed below.

1.      I couldn’t find any information about the X’tal structure deposition in the main text of the manuscript. Are those two crystal-structures deposited to the protein data bank? If yes, then the validation report should be provided along with a revised version. If no, then they should be deposited and deposition ids should be included in the revised manuscript. Well, there are pdb ids mentioned in Supplementary table S1, but the mentioned pdb id, 8AGI seems to lead to a different entry, not to the OXO40-DF004-mab structure. Please verify it.

2.      The data collection and refinement statistics: The Ramachandran plot preferred, allowed and outliers are not mentioned properly in both structures. First, Ramachandran allowed residue for OXO40/DF004-Fab structure is missing. Second, the total of preferred, allowed and outlier residues (%) is not 100%; should be 100%.

3.      Atomic coordinates (pdb files), electron density (mtz files) and validation reports for both structures should be provided for better assessment of the structural results.

4. Statistical deviations (error bars) in Fig. 2F, 2G, 3D and 3E should be mentioned.  

5.  The language of the manuscript can be improved. 
